# HEART Score and Its Implementation in Emergency Medicine Departments in the West Balkan Region—A Pilot Study

**DOI:** 10.3390/healthcare11172372

**Published:** 2023-08-23

**Authors:** Armin Šljivo, Ahmed Mulać, Amina Džidić-Krivić, Katarina Ivanović, Dragana Radoičić, Amina Selimović, Arian Abdulkhaliq, Nejra Selak, Ilma Dadić, Stefan Veljković, Slobodan Tomić, Leopold Valerian Reiter, Zorana Kovačević, Sanja Tomić

**Affiliations:** 1Clinical Center of University of Sarajevo, 71000 Sarajevo, Bosnia and Herzegovina; 2Cantonal Hospital Zenica, 72000 Zenica, Bosnia and Herzegovina; 3Faculty of Medicine, University of Belgrade, 11000 Belgrade, Serbia; 4Institute for Cardiovascular Disease Dedinje, 11000 Belgrade, Serbia; 5Faculty of Medicine, Iuliu Haţieganu University of Medicine and Pharmacy Cluj-Napoca, 400012 Cluj-Napoca, Romania; 6Dom Zdravlja Zenica, 72000 Zenica, Bosnia and Herzegovina; 7Dom Zdravlja Sarajevo, 71000 Sarajevo, Bosnia and Herzegovina; 8Faculty of Medicine, University of Novi Sad, 21000 Novi Sad, Serbiasanja.tomic@mf.uns.ac.rs (S.T.)

**Keywords:** angina pectoris, myocardial infarction, HEART score, emergency department, risk score

## Abstract

Background: Chest pain represents a prevalent complaint in emergency departments (EDs), where the precise differentiation between acute coronary syndrome and alternative conditions assumes paramount significance. This pilot study aimed to assess the HEART score’s implementation in West Balkan EDs. Methods: A retrospective analysis was performed on a prospective cohort comprising patients presenting with chest pain admitted to EDs in Sarajevo, Zenica, and Belgrade between July and December 2022. Results: A total of 303 patients were included, with 128 classified as low-risk based on the HEART score and 175 classified as moderate-to-high-risk. The low-risk patients exhibited younger age and a lower prevalence of cardiovascular risk factors. Laboratory and anamnestic findings revealed higher levels of C-reactive protein, ALT, and creatinine, higher rates of moderately to highly suspicious chest pain history, a greater number of cardiovascular risk factors, and elevated troponin levels in moderate-to-high-risk patients. Comparatively, among patients with a low HEART score, 2.3% experienced MACE, whereas those with a moderate-to high-risk HEART score had a MACE rate of 10.2%. A moderate-to-high-risk HEART score demonstrated a sensitivity of 91.2% (95%CI 90.2–93.4%) and specificity of 46.5% (95%CI 39.9–48.3%) for predicting MACE. Conclusion: This pilot study offers preliminary insights into the integration of the HEART score within the emergency departments of the West Balkan region.

## 1. Introduction 

Chest pain is a ubiquitous health complaint encountered by healthcare providers across various medical specialties. It presents as a feeling of pressure or heaviness in the chest area or discomfort which can vary in intensity and duration and can be associated with various underlying causes, including cardiac and non-cardiac conditions [1]. Intriguingly, chest pain is the second most commonly presented symptom in US emergency departments, with approximately 80% of patients seeking care eventually being diagnosed without acute coronary disease. Only one-third of the remaining 20% receive a diagnosis of and treatment for acute coronary syndrome (ACS), encompassing a spectrum ranging from unstable angina pectoris to acute myocardial infarction (AMI) [2,3]. There is an extreme lack of information concerning the admission of patients with chest pain to emergency departments in the West Balkan region. As a result, there is a complete absence of both official and unofficial data regarding these admissions in our countries.

In the setting of chest pain, the clinician’s objective is to distinguish between patients who exhibit symptoms of ACS and those who have other, usually less harmful conditions. While several historical features and laboratory results can be valuable in identifying patients with authentic ACS, their reliability becomes limited when dealing with individuals who have concurrent comorbidities, such as chronic kidney disease, systemic inflammation, viral myocarditis, other cardiac diseases, infiltrative diseases (e.g., sarcoidosis), or even acute stroke. Consequently, none of these indicators can be considered entirely precise or independently reliable in such complex cases, and in these patients, the utilization of a risk score becomes essential, as it can serve as a valuable tool to predict the likelihood of developing ACS in the future [4]. Consequently, each year, approximately 2% to 5% of patients with actual ACS are mistakenly discharged from emergency departments, and overlooking cases of ACS constitutes a substantial portion of medical malpractice claims [5]. In light of this information, there is a tendency to engage in excessive testing and invasive diagnostic procedures for chest pain patients, even in those deemed at low risk. This inclination contributes to increased healthcare expenditures without demonstrating significant enhancements in clinical outcomes [6,7].

Several decision instruments are currently available to identify chest pain patients at low risk who are eligible for discharge without further testing. These include the Thrombolysis in Myocardial Infarction score (TIMI), the Emergency Department Assessment of Chest Pain score (EDACS) and the History, ECG, Age, Risk factors and Troponin (HEART) score. The TIMI score evaluates the 14-day risk of major adverse cardiac events (MACE). Although the American Heart Association and the American College of Cardiology have endorsed the use of the TIMI score for the initial evaluation of chest pain patients, its applicability in identifying low-risk individuals has yielded conflicting findings [8]. On the other hand, the EDACS score demonstrated high sensitivity in identifying nearly half of the patients who arrived at emergency departments with potential cardiac chest pain as having a low risk of experiencing major adverse cardiac events in the short term [9].

Lastly, the HEART [10] score is one of the most important tools that is being used in clinical practice for ACS evaluation and risk stratification, because it is very easy to apply and its main focus includes short-term outcomes, which is an especially significant factor to consider when working in the emergency department. In the assessment of chest pain patients, the HEART score exhibited higher discriminative capability in identifying individuals with and without major adverse cardiac events (MACE) when compared to the GRACE and TIMI scores. Additionally, the HEART score identified a substantial cohort of low-risk patients while maintaining a comparable level of safety in risk stratification [11]. A HEART score ranging from 0 to 3 indicates that a patient is classified as “low risk” for MACE and may be considered for discharge. Patients with a score of 4 to 6 are categorized as “moderate risk”, while those with a score of 7 to 10 are classified as “high risk”. For patients with a moderate-risk HEART score, admission for additional testing is recommended, while those with a high risk should be evaluated for potential invasive testing. The HEART score has undergone independent validation in various studies globally since its initial validation [10,11,12,13]. To better comprehend this tool’s overall prognostic accuracy and its potential role in assessing the risk of patients experiencing chest pain, further research is necessary.

Our study was undertaken with the primary objective of introducing the HEART score, a widely recognized risk stratification tool, and subsequently implementing the HEART pathway in the emergency departments of the West Balkans region, with the future plan being to collect information regarding the burden of chest pain entities, heart disease morbidity, mortality and HEART score outcomes. This pilot study presents the initial results concerning the utilization of the HEART score, the predicting of adverse cardiac events within 30 days, as well as the implementation of this diagnostic tool in routine clinical practice. By introducing the HEART score and pathway, our study sought to address the need for standardized risk stratification and management approaches for patients presenting with chest pain in the emergency setting. The HEART score, a validated risk assessment tool, offers a systematic approach to evaluating patients based on their historical, clinical, and electrocardiographic characteristics, as well as troponin measurements. The HEART pathway then guides subsequent diagnostic and therapeutic interventions based on the calculated risk level.

## 2. Materials and Methods

This study conducted a retrospective analysis of a prospective cohort consisting of patients who were admitted to the emergency medicine departments of Sarajevo, Zenica, and Belgrade during the period from July to December 2022. The timeframe of this study coincided with the ongoing COVID-19 pandemic, which added an additional layer of complexity to the management of patients presenting with various medical conditions, including chest pain. To ensure adherence to ethical standards and appropriate research protocols, the study received approval from the Bioethical Committee of Dom Zdravlja Zenica (OU-01.1-99-40435/22). The research was conducted in accordance with all the relevant amendments of the Helsinki Declaration, which outlines ethical guidelines for medical research involving human subjects. This ensured the protection of patient rights, privacy, and confidentiality throughout the study.

### 2.1. Subjects

The subjects included in the study setting were patients referred to the emergency medicine departments due to chest pain in Sarajevo, Zenica and Belgrade. The exclusion criteria consisted of the following: (i) patients presenting with ST elevation myocardial infarction; (ii) patients who left the emergency department against medical advice; (iii) patients who declined to give consent for the research; (iv) patients who passed away in the emergency department; and (v) patients who were transferred to other regional hospitals. Prior to their participation in the study, all participants received comprehensive information regarding the study’s objectives. They were informed about the voluntary nature of their participation and were required to provide informed consent. Furthermore, patients were provided with details regarding the specific data collected for the study purposes.

### 2.2. Study Instruments and Data Collection

Clinical data were collected using a standardized case report form in accordance with established clinical data standards. Trained clinicians, later-year residents or specialists in emergency medicine, internal medicine, or cardiology, and medical doctors with one year of clinical experience gathered all of the components of the HEART score, including patient history, ECG, age, risk factors and troponin levels. Following the evaluation in the emergency department (ED), the disposition of the patients was categorized as either discharged or undischarged, which included cases where patients were hospitalized or referred to a cardiologist. During the 30-day follow-up period, data were collected from both digital and written patient records, including discharge letters, revascularization reports, and other relevant documentation. In instances where follow-up data were not accessible in the hospital records, proactive measures such as direct communication with the patient or with their general practitioner were undertaken to establish communication, in order to procure comprehensive information pertaining to the patient’s medical status, hospital admissions, occurrences of myocardial infarction, and revascularization procedures. MACE was defined as a composite outcome that included death, AMI, and revascularization, either through percutaneous coronary intervention (PCI) or coronary artery bypass grafting (CABG) [14]. The 30-day follow-up period is considered relevant for capturing immediate post-acute events and complications that might occur shortly after an index event or intervention. This helps one to understand the immediate impact of the treatment or intervention being studied. Furthermore, using a longer follow-up period may introduce potential biases due to loss to follow-up, changes in treatment strategies, or unrelated events that could confound the study outcomes. A shorter follow-up period minimizes the risk of these potential confounding factors and ensures more focused and accurate data collection.

The HEART score [10], incorporating patient history, ECG, age, risk factors and troponin levels, ranges from 0 to 10 and is calculated by summing up all the components of the score, with scores between 0 and 3 being classified as low-risk, scores between 4 and 6 being classified as moderate-risk, and scores between 7 and 10 being classified as high-risk. The absence of specific details regarding chest pain, such as its pattern, onset, duration, relationship to exercise, stress or cold, location, accompanying symptoms, and response to sublingual nitrates, resulted in a medical history being labeled as “non-specific” with zero points, while a combination of non-specific and suspicious elements led to a classification of “moderately suspicious” with one point, and primarily specific elements indicated a “highly suspicious” status with two points.

The ECG was categorized as follows; ”normal” ECG based on the Minnesota criteria was assigned zero points; in cases with repolarization abnormalities without notable ST-segment depression, the presence of a bundle branch block, typical abnormalities suggesting left ventricular hypertrophy, repolarization abnormalities likely due to digoxin use, or if known repolarization disturbances remained unchanged, one point was assigned; and two points were assigned for significant ST-segment depressions or elevations in the absence of a bundle branch block, left ventricular hypertrophy, or digoxin use.

The patient’s age at the time of admission was taken into consideration. If the patient was younger than 45 years, zero points were assigned. If the patient’s age fell between 45 and 65 years, one point was given. In cases where the patient was 65 years or older, two points were awarded.

The assessment of risk factors for coronary artery disease involved counting the number of factors, including currently treated diabetes mellitus, current or recent smoking, diagnosed hypertension, diagnosed hypercholesterolemia, a family history of coronary artery disease, and obesity. If none of these risk factors were present, zero points were assigned; one or two risk factors resulted in one point; and three or more risk factors were awarded two points. Additionally, a history of coronary revascularization, myocardial infarction, stroke, or peripheral arterial disease resulted in two points.

Based on the troponin I level which was assessed upon arrival, points were assigned, with zero points for levels below or equal to the positivity threshold (≤0.04), one point for levels between one and two times the threshold, and two points for levels exceeding two times the threshold. There were no distinctions based on gender in terms of the reference values for troponin levels in our laboratories.

Following the initial evaluation, we divided the patients into two categories: those with a low-risk HEART score and those with a moderate-to-high-risk HEART score. Patients with a moderate-to-high-risk HEART score were directed either for observation or towards further cardiological tests, both invasive and noninvasive, to identify the cause of the chest pain—whether it was related to cardiac ischemia, non-ischemic cardiac issues, or if the pain imitated other medical conditions.

### 2.3. Statistical Analysis

The collected data were analyzed and summarized using descriptive statistics. For normally distributed data, results were presented as frequencies and percentages along with the mean ± standard deviation. Non-normally distributed data were reported as the median accompanied by 25th and 75th percentiles. To assess the relationship between different variables and specific phenomena, appropriate statistical tests were employed, including independent samples *t*-tests, Mann–Whitney U tests, or chi-square tests. The statistical significance level was set at *p* < 0.05 for all tests, and a two-sided approach was used.

## 3. Results

In this pilot study, a total of 303 patients were included in the study setting, after excluding 15 cases because of the exclusion criteria. Our population was predominately male (173, 57.1%), with a mean age of 64.0 ± 13.4, and 239 participants presented with hypertension (78.9%), 83 presented with diabetes mellitus (27.4%), 209 presented with hyperlipidemia (68.9%), 130 had a BMI >25 kg/m^2^ (42.9%), and 101 (33.3%) were active smokers. The HEART score mostly revealed moderately suspicious patient history (121, 39.9%), normal or nonspecific ECG changes (159, 52.4%), and age group of 45–65 years (143, 47.1%), the presence of 3 and more risk factors (156, 51.5%) and troponin levels within the normal range (235, 77.5%). Within 30 days of admission to the emergency department, out of the 303 patients, 34 (11.3%) were diagnosed with MACE, including AMI (13, 4.2%), 4 died (1.3%), 5 (1.6%) had a CABG and 12 (3.9%) underwent PCI.

The sample included 128 (42.2%) patients with a low-risk HEART score and 175 (57.8%) with a moderate-to-high-risk HEART score. Patients with a low-risk HEART score were significantly (*p* = 0.002) younger (60.7 ± 14.8 vs. 67.3 ± 12.1) and had a lower incidence of cardiovascular risk factors when compared to moderate-to high-risk HEART score patients. All other demographic data (sex, age) and cardiovascular-associated risk factors are presented in Table 1. 

All patients included in the study were admitted to the emergency departments because of chest pain and after taking anamnesis, and the information regarding the quality of the chest pain was referred to for laboratory analysis. The initial laboratory revealed that moderate-to-high-risk HEART score patients had a significantly (*p* < 0.05) higher mean score of C-reactive protein (22.0 ± 3.8 vs. 7.0 ± 4.1) and creatinine (118.0 ± 56.8 vs. 83.4 ± 35.1) when compared to low-risk HEART score patients. There was no significant (*p* > 0.05) difference in the mean systolic and diastolic pressure between the low-risk HEART score and moderate-to-high-risk HEART score patient groups. All other laboratory, blood pressure and ECG findings among low-risk HEART and moderate-to-high-risk HEART score patients are presented in Table 2.

Out of the whole study sample, 68 (22.4%) patients were admitted for further cardiovascular investigation (echocardiography, coronary angiography, etc.). Among them, 44 (14.5%) had AMI, 12 (3.9%) had unstable angina pectoris, 7 (2.3%) had arrythmia, and 5 (1.6%) had other cardiac diseases. Patients with a moderate-to-high-risk HEART score had significantly higher rates of moderately to highly suspicious chest pain history, more cardiovascular risk factors, and higher troponin levels (*p* < 0.001). The moderate-to-high-risk HEART score (≥4) for predicting AMI had a sensitivity of 91.2% (95% CI 78.1–93.3%) and a specificity of 51.9% (95% CI 35.8–58.2%). Please refer to Table 3 for additional information regarding patient history, ECG findings, age, risk factors, and troponin levels for both low-risk and moderate-to-high-risk HEART score patients.

Within 30 days of admission to the emergency department, out of the 303 patients, 34 (11.3%) were diagnosed with MACE, including AMI (13, 4.2%), 4 died (1.3%), 5 (1.6%) had a CABG and 12 (3.9%) underwent PCI. Among patients with low HEART score, 2.3% (3/128, with 2 AMI and 1 PCI) had MACE compared to 10.2% of those with a moderate-to-high-risk HEART score (31/175, with 11 AMI, 4 deaths, 5 had a CABG and 12 underwent PCI). A moderate-to-high-risk HEART score had a sensitivity of 91.2% (95% CI 90.2–93.4%) and a specificity of 46.5% (95% CI 39.9–48.3%) for MACE. The four deaths observed in this study were of cardiac cause and were in the high-risk HEART group. 

## 4. Discussion

To our knowledge, this study is the first in the West Balkans to assess the usage of a standardized HEART protocol for evaluating chest pain symptoms and their association with 30-day adverse cardiac events. It highlights the significance of introducing the HEART pathway in emergency departments. Our study sample primarily consisted of male patients who were older, obese, and had hypertension and hyperlipidemia. Patients with low-risk HEART scores were younger and had a lower incidence of cardiovascular risk factors compared to those with moderate-to-high-risk scores. Conversely, patients with moderate-to-high-risk scores exhibited higher rates of suspicious chest pain history and elevated troponin levels. Also, a moderate-to-high-risk HEART score had high sensitivity but rather low specificity for MACE.

Regarding demographic data, the low-risk HEART score group had a younger mean age, consistent with previous studies [15], which suggests that age plays a significant role in determining the risk level for MACE. These findings underscore the effectiveness of the HEART score in age-related as well as risk factor-related terms as a comprehensive risk assessment tool. By incorporating information about hypertension, diabetes, hyperlipidemia, smoking, and family history of cardiovascular disease, the HEART score captures the impact of these risk factors on the overall risk evaluation. It provides a systematic approach to quantifying and evaluating the presence and significance of these factors in cardiac patients. Consequently, the HEART score enhances the accuracy of risk stratification, enabling healthcare professionals to make informed decisions regarding patient management and treatment strategies.

The laboratory findings in this study further support the discriminative ability of the HEART score. Patients with moderate-to-high-risk HEART scores exhibited significantly higher levels of C-reactive protein and creatinine compared to low-risk patients. These biomarkers have been linked to inflammatory processes and renal impairment, which are commonly associated with acute coronary syndrome and cardiovascular diseases [16,17]. The elevated creatinine levels observed in the moderate-to-high-risk group further strengthen the association between impaired renal function, an elevated HEART score and potential or proven acute coronary syndrome. Approximately 30–40% of patients diagnosed with non-ST-segment elevation acute coronary syndrome exhibit renal dysfunction [18,19]. The elevated levels of these biomarkers in the moderate-to-high-risk group indicate a higher likelihood of underlying cardiac pathology and support the utility of the HEART score in identifying patients who may require further cardiovascular investigations.

Comparisons with other studies reinforce the effectiveness of the HEART score in risk stratification. Previous research by Backus et al. [20] and Poldervaart et al. [21] demonstrated the higher sensitivity and negative predictive value of the HEART score compared to physician judgment, indicating its reliability in identifying low-risk patients and potentially reducing unnecessary hospitalizations. These findings have important implications for clinical practice. By utilizing the HEART score as a risk assessment tool, healthcare professionals can make more informed decisions regarding the appropriate level of care for patients presenting with potential cardiac symptoms. The higher sensitivity and negative predictive value of the HEART score make it a valuable tool for safely triaging low-risk patients to outpatient settings, reducing healthcare costs and optimizing resource allocation. Similarly, a study by Body et al. [22] highlighted the superiority of the HEART score over other risk assessment tools, such as the TIMI and GRACE scores, in terms of sensitivity and negative predictive value. The low specificity of the HEART score can be attributed to several factors. The HEART score includes symptoms such as chest pain or discomfort, which can also be present in various non-cardiac conditions [23]. As a result, some patients without significant cardiac issues might still score higher on the HEART score due to the presence of non-specific symptoms, leading to lower specificity. The assessment of history and risk factors in the HEART score relies on subjective judgments by healthcare providers. Different clinicians might interpret and weigh these factors differently, potentially impacting the accuracy of the score and contributing to lower specificity. Elevated troponin levels are an essential component of the HEART score. However, troponin levels can also be elevated in conditions other than acute coronary syndrome (ACS), such as myocarditis or pulmonary embolism [24]. This lack of specificity in troponin elevation can lead to a reduction in the overall specificity of the HEART score. The HEART score exhibits a higher level of predictive accuracy for cardiovascular events within the 30-day follow-up period when compared to alternative scoring methodologies such as GRACE and TIMI scores. The integration of high-sensitivity troponins sustains the demonstrated superiority of this score, thereby facilitating a more precise identification of low-risk patients [25].

While the HEART score has demonstrated its effectiveness in risk stratification, it is essential to recognize that it should be utilized as one component of a comprehensive clinical evaluation. The assessment of cardiac risk should involve a combination of various factors, including patient history, physical examination, and additional diagnostic tools such as electrocardiography and cardiac biomarkers. The HEART score serves as a valuable tool in providing risk stratification information, but it should not be solely relied upon in isolation. Clinical judgment plays a crucial role in the overall evaluation and management of patients. Physicians must consider individual patient characteristics, medical history, and specific clinical circumstances when interpreting the results of the HEART score. Each patient is unique, and factors such as comorbidities, atypical symptoms, or other relevant clinical findings may influence the risk assessment and subsequent management decisions. The integration of the HEART score into clinical practice allows healthcare professionals to adopt a standardized and structured approach to risk stratification. However, it is important to remember that the HEART score is not a substitute for clinical judgment. It serves as a valuable tool in guiding decision-making, but it should be used in conjunction with a thorough assessment of the patient’s overall clinical picture.

Initially, there were some challenges encountered by institutions interested in integrating the HEART score into their practice. Institutions faced challenges in clinician acceptance and adoption of the HEART score due to concerns about its accuracy and potential behavioral changes required. To address this, a multi-pronged approach was employed. Workshops and seminars were organized, led by experienced clinicians and experts in cardiovascular risk assessment. These sessions provided in-depth training on the proper application of the HEART score, including the interpretation of its components (history, ECG, age, risk factors, and troponin) and the risk categories. Successfully integrating the HEART score into the clinical workflow required collaboration between clinical teams and the institution’s information technology department. User-friendly electronic decision support systems linked directly to the electronic health records were used in the emergency departments. This software allowed clinicians to input patient data, such as medical history, ECG findings, age, and risk factors, which were then processed to calculate the HEART score. We used visual aids and multimedia resources to simplify complex medical terms and ensure that patients had a clear understanding of their risk assessment and its implications. Patients’ questions and concerns were addressed with empathy, fostering trust and confidence in the use of the HEART score as part of their care. Institutions implemented monitoring and quality improvement initiatives to ensure ongoing compliance and adherence to using the HEART score. Regular audits were conducted to assess how consistently the HEART score was applied across different clinical cases.

Despite the strengths of this study and the supporting evidence from previous research, there are some limitations to consider. The current study being examined is a pilot study, which means that it is a preliminary investigation designed to test the feasibility, methods, and data collection procedures before conducting a more comprehensive and definitive research project. While pilot studies provide valuable insights, they usually involve a smaller sample size and are limited to specific locations, which might not fully represent the diversity and complexity of the target population. Future studies will include a significantly larger number of participants compared to this pilot study, and efforts will be made to recruit participants from various geographical locations, representing diverse communities, cultures, and backgrounds. This will help to capture regional variations and ensure that the study’s conclusions are not limited to a specific group but can be extended to a more diverse population. The research also did not take into account any age-related factors in relation to the laboratory results, and there is a possibility that certain medications used by patients might have impacted the laboratory findings. Additionally, the study was conducted in a specific setting and may not reflect the diversity of patient populations and healthcare systems in other regions. Further research involving larger cohorts and diverse patient populations is warranted to validate the findings and assess the applicability of the HEART score in different clinical contexts. Lastly, the applicability of the HEART score is primarily relevant to women experiencing chest pain with typical symptoms, given that women frequently present with atypical symptoms. Consequently, this divergence in symptom presentation among women could create a gap in the effectiveness and applicability of the HEART score for this subgroup. As the score’s design centers around specific symptomatology, its predictive accuracy might be compromised in cases where women present with symptoms that are not congruent with the established norm. Our future research will also try to incorporate gender differences which contribute to the cardiac disease burden due to the mentioned criteria.

## 5. Conclusions

In conclusion, the findings of this pilot study align with previous research, confirming the utility of the HEART score in risk stratification for patients with chest pain. By considering factors such as age, cardiovascular risk factors, and biomarker levels, the HEART score accurately assesses the overall risk profile of patients, allowing for appropriate risk stratification. The HEART score effectively differentiates patients into low-risk and moderate-to-high-risk categories, based on their demographic characteristics and laboratory findings. Comparisons with other studies consistently demonstrate the superior sensitivity and negative predictive value of the HEART score compared to other risk assessment tools. Implementing the HEART score as part of the clinical decision-making process offers several advantages. It aids in the appropriate triage and management of patients by providing a standardized and objective approach to risk assessment. By incorporating the HEART score into clinical practice, healthcare professionals can make more informed decisions regarding the level of care required for individual patients, optimizing resource allocation and improving patient outcomes.

## Figures and Tables

**Table 1 healthcare-11-02372-t001:** Sex, age and cardiovascular-associated risk factors such as hypertension, diabetes mellitus, hyperlipidemia, smoking status and a positive family history of cardiovascular disease among low-risk HEART and moderate-to-high-risk HEART score patients admitted to the emergency medicine departments of the West Balkan region.

		Low-Risk HEART Score (3 ≤ Points) N = 128	Moderate-to-High-Risk HEART Score (>3 Points) N = 175
Sex (No, %)	Male	90 (70.3%)	83 (47.4%)
	Female	38 (29.7%)	92 (52.6%)
	Mean age	60.7 ± 14.8	67.3 ± 12.1
Risk factors (No, %)	Hypertension	89 (69.5%)	150 (85.7%)
	Diabetes mellitus	8 (6.2%)	75 (42.8%)
	Hyperlipidemia	73 (57.0%)	136 (77.7%)
	Smoking	43 (33.5%)	58 (33.1%)
	BMI > 25 kg/m^2^	39 (30.4%)	91 (52.0%)
	Family history	57 (44.5%)	96 (54.8%)

**Table 2 healthcare-11-02372-t002:** Laboratory, blood pressure and ECG findings among low-risk HEART and moderate-to-high-risk HEART score patients admitted to the emergency medicine departments of West Balkan.

Parameters		Low-Risk HEART Score (3 ≤ Points) N = 128	Moderate-to-High-Risk HEART Score (>3 Points) N = 175	*p*-Value
Laboratory findings (mean ± SD)	C-reactive protein	7.0 ± 4.1	22.0 ± 3.8	<0.05
	D-dimer	0.9 ± 0.5	1.1 ± 0.6	>0.05
	AST	47.0 ± 5.3	66.5 ± 6.2	>0.05
	ALT	36.5 ± 19.9	42.38 ± 17.2	<0.05
	CK	57.6 ± 13.2	75.5 ± 8.5	>0.05
	CKMB	12.4 ± 5.3	20.0 ± 7.1	>0.05
	LDH	323.3 ± 26.8	431.6 ± 31.0	>0.05
	Urea	7.5 ± 6.2	10.7 ± 9.3	>0.05
	Creatinine	83.4 ± 35.1	118.0 ± 56.8	<0.05
Blood pressure (mean ± SD)	Systolic	132.7 ± 16.9	138.4 ± 21.8	>0.05
	Diastolic	81.8 ± 8.8	83.5 ± 10.9	>0.05
ECG findings (No, %)	Normal or nonspecific	117 (91.4%)	42 (24.0%)	N/A
	Nonspecific repolarization disturbance (LBBB or inverted T waves)	11 (9.6%)	69 (39.4%)	N/A
	Significant ST depression	0 (0%)	64 (36.6%)	N/A

AST—aspartate transaminase; ALT—alanine transaminase; CK—creatin kinase; CKMB—creatine kinase fraction MB; LDH—lactate dehydrogenase; LBBB—left bundle branch block.

**Table 3 healthcare-11-02372-t003:** Comparison of HEART score and its components between low-risk HEART and moderate-to-high-risk HEART score patients.

			Low-Risk HEART Score (3 ≤ Points) N = 128	Moderate-to-High-Risk HEART Score (>3 Points) N = 175	*p*-Value
HEART SCORE	Patient history	Slightly suspicions	82 (64.1%)	37 (21.1%)	<0.001
		Moderately suspicions	37 (28.9%)	84 (48.0%)	
		Highly suspicious	9 (7.1%)	54 (30.9%)	
	ECG	Normal or nonspecific	117 (91.4%)	42 (24.0%)	N/A
		Nonspecific repolarization disturbance (LBBB or inverted T waves)	11 (9.6%)	69 (39.4%)	
		Significant ST depression	0 (0%)	64 (36.6%)	
	Age	<45 years	27 (21.1%)	3 (1.7%)	0.002
		45–65 years	81 (63.3%)	62 (35.4%)	
		>65 years	20 (15.6%)	110 (62.9%)	
	Risk factors	No risk facrtors known	20 (15.6%)	2 (1.1%)	<0.001
		1 or 2 risk factors	87 (67.9%)	38 (21.7%)	
		>3 risk factors/history of atherosclerosis disease	21 (16.5%)	135 (77.2%)	
	Troponin	<normal limit	122 (95.3%)	113 (64.6%)	<0.001
		1–2x normal unit	4 (3.1%)	30 (17.1%)	
		>2x normal limit	2 (1.6%)	32 (18.3%)	

## Data Availability

The data are available from the authors on personal request.

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
