# Peer review of "HEART Score and Its Implementation in Emergency Medicine Departments in the West Balkan Region—A Pilot Study"

_healthcare, 2023, doi:10.3390/healthcare11172372_

Round 1
Reviewer 1 Report
The paper presents a pilot study to assess the implementation and effectiveness of the HEART score in the West Balkan region emergency departments. Using this validated risk stratification tool, the study aims to distinguish between patients with acute coronary syndrome (ACS) and those with other conditions. The report includes a retrospective analysis of clinical data from patients presenting with chest pain and calculates their HEART scores, determining their risk level. The study offers initial insights into how the HEART score can identify patients at low risk of experiencing adverse cardiac events using a sample size of 303 patients.
Strengths:
1. The study's aim is clearly defined and relevant, addressing a common and clinically significant problem: differentiating between ACS and other conditions in patients presenting with chest pain.
2. Using the validated HEART score as a risk stratification tool adds credibility to the study.
3. The study design is sound, with a clear description of the methods used for data collection, including patient history, ECG results, age, risk factors, and troponin levels.
4. Including various data points (clinical, demographic, laboratory) offers a comprehensive view of each patient's health status.
5. The statistical analysis appears robust, with appropriate use of significance testing and confidence intervals.
6. The research has immediate clinical applicability, potentially helping physicians make more informed decisions about patient care.
Drawbacks:
1. The sample size is relatively small (303 patients) for a tool designed to predict clinical outcomes, reducing the generalizability of the results.
2. The study only includes patients from three emergency departments in the West Balkan region, limiting its applicability to a broader population or other geographical regions.
3. The study is a retrospective analysis of a prospective cohort, which could be prone to information bias or loss of follow-up.
4. The specificity of the HEART score (46.5%) is quite low, indicating a relatively high rate of false positives.
5. There is no mention of the implementation challenges and how they were overcome.
6. The study lacks a comparison with other risk stratification tools or standard care in these settings.
Recommendations:
1. Increase the sample size and diversify the geographical locations to enhance the generalizability of the study.
2. Perform a prospective study to avoid potential bias and ensure all relevant and necessary data is collected appropriately.
3. Explore the reasons behind the relatively low specificity of the HEART score and look for ways to improve it or discuss its implications.
4. Include a comparison group where other risk stratification tools are used or standard care is provided to provide a point of reference and ascertain the HEART score's added value.
5. Elaborate on implementation challenges and how they were addressed, which can offer valuable lessons for other institutions interested in using the HEART score.
6. Given the low specificity, consider using the HEART score with other diagnostic tools rather than in isolation.
7. Future studies could evaluate the long-term impact of the HEART score on patient outcomes and healthcare resource utilization.
8. The paper should be expanded. It is recommended to expand the current analysis section and the discussion section.
In conclusion, the study provides preliminary evidence suggesting the potential benefits of integrating the HEART score in emergency departments in the West Balkan region. However, a more comprehensive and more extensive scale study, perhaps in different regions and settings, will be beneficial in reinforcing these initial findings and understanding the overall implications better.
Author Response
Dear Reviewer,
thank You very much for an extensive overview of the paper submitted to Healthcare. According to the recommendation we have corrected the paper and answered all points.
- Increase the sample size and diversify the geographical locations to enhance the generalizability of the study.
The current study being examined is a pilot study, which means it is a preliminary investigation designed to test the feasibility, methods, and data collection procedures before conducting a more comprehensive and definitive research project. While the pilot study provides valuable insights, it usually involves a smaller sample size and is limited to specific locations, which might not fully represent the diversity and complexity of the target population.
The future study will include a significantly larger number of participants compared to the pilot study. By increasing the sample size, the paper will have more statistical power to detect meaningful relationships and draw more reliable conclusions about the variables under investigation. In the future study, efforts will be made to recruit participants from various geographical locations, representing diverse communities, cultures, and backgrounds. This will help capture regional variations and ensure that the study's conclusions are not limited to a specific group but can be extended to a more diverse population.
We have added this notice in the limitations section of the DIscussion.
- Perform a prospective study to avoid potential bias and ensure all relevant and necessary data is collected appropriately.
Performing a prospective study is a crucial step in the research process, especially when aiming to fully undestand and show the implication of the HEART score. We again point out that this is only a pilot study, which test the feasibility, methods, and data collection procedures before conducting a more comprehensive and definitive research project
- Explore the reasons behind the relatively low specificity of the HEART score and look for ways to improve it or discuss its implications.
We have explained it in the DIscussion section.
- Include a comparison group where other risk stratification tools are used or standard care is provided to provide a point of reference and ascertain the HEART score's added value.
Future study will include groups of patients.
- Elaborate on implementation challenges and how they were addressed, which can offer valuable lessons for other institutions interested in using the HEART score.
We have added this point in the text.
- Given the low specificity, consider using the HEART score with other diagnostic tools rather than in isolation.
We have explained it in the Discussion section.
- Future studies could evaluate the long-term impact of the HEART score on patient outcomes and healthcare resource utilization.
We aim to conduct a 5 year long term risk stratification of the HEART score, after this iniital pilot study which will include patients outcomes and even finantial burden to the healthcare system.
- The paper should be expanded. It is recommended to expand the current analysis section and the discussion section.
We have expanded the paper as recommended.
Reviewer 2 Report
Chest pain is a common presenting complaint in emergency departments, and accurately distinguishing between patients with acute coronary syndrome (ACS) and those with other conditions is crucial.
The authors proposed a pilot study aimed to assess the implementation of the HEART score in emergency departments in the West Balkan.
They performed retrospective analysis on a prospective cohort comprising patients presenting with chest pain who were admitted to the emergency departments in Sarajevo, Zenica, and Belgrade between July and December 2022.
They collected systematically clinical data, encompassing patient history, electrocardiogram (ECG) results, age, risk factors, and troponin levels, were systematically collected.
The HEART score, a validated risk stratification tool, was calculated for each patient, enabling their categorization into low-risk, moderate-risk, or high-risk groups. A total of 303 patients were included, with 128 classified as low-risk based on the HEART score and 175 as moderate- to high-risk. The low- risk patients exhibited significantly younger age and a lower prevalence of cardiovascular risk factors. Laboratory findings revealed significantly higher levels of C-reactive protein, ALT, and creatinine in patients classified as moderate-to high-risk. Additionally, patients with moderate- to high-risk HEART score exhibited significantly (p<0.001) higher rates of moderately to highly suspicious chest pain history, greater number of cardiovascular risk factors and elevated troponin levels. Comparatively, among patients with a low HEART score, 2.3% experienced MACE, whereas those with a moderate- to high-risk HEART score had a MACE rate of 10.2%. The moderate- to high-risk HEART score demonstrated a sensitivity of 91.2% (95% CI 90.2%–93.4%) and specificity of 46.5% (95% CI 39.9%–48.3%) for predicting MACE.
Overall the authors offer preliminary insights into the integration of the HEART score within the emergency departments of the West Balkan region. The findings indicate that the HEART score demonstrates effectiveness in identifying patients with a low risk of experiencing adverse cardiac events.
This is an interesting piece.
Some minor comments for the authors:
1. There are a lot of acronyms. Minimize and make a list.
2. The aim “Our study aimed to introduce the HEART score and subsequently the HEART pathway in the emergency departments of the West Balkans” must be more effective.
3. “The HEART score” in the methods is a subparagraph.
4. Smooth and improve the results. I suggest, first to summarize them and then to arrange them in to themes-paragraphs.
Author Response
Dear Reviewer,
thank You for Your extensive review. We have corrected all the points as mentioned in the comments. According to them here are our responses.
- There are a lot of acronyms. Minimize and make a list.
The list has been made and added to the manuscript.
- The aim “Our study aimed to introduce the HEART score and subsequently the HEART pathway in the emergency departments of the West Balkans” must be more effective.
We have rephrased it to be more effective.
- “The HEART score” in the methods is a subparagraph.
We have removed the subtitle The Heart score.
- Smooth and improve the results. I suggest, first to summarize them and then to arrange them in to themes-paragraphs.
We havve made the results more smooth and improved them.
Reviewer 3 Report
Dear Authors
I enjoyed reading the manuscript as well as the proposal for the implementation of the HEART in the European region of its setting. I hope the authors get success in their goals.
Although "an easy to obtain data study" there is originality in its results, as mentioned. I have a question due to the following results presented in "Table 1" and I would like the authors to review carefully these preliminary presented results to see if they are really correct and if the numbers are trustworthy. Additionally, in the discussion, I suggest quoting your results with the results of previous publications that come from other countries, and other populations.
In my view, the discussion seems lengthy and I would point to make it more concise and also include more information from previous publications.
As a pilot study, the reviewer encourages the authors to pursue this research field considering that this may arise the interest of other groups also contribute.
Regards,
Although the Editor of the periodical will certainly count on English reviewing I point out that is always of utmost importance to rely on the contribution of native English-speaking reviewers to approve the final writing of the manuscript.
Author Response
Dear Editor,
thank You very much for such words. We have corrected all the points that You have mentioned and rechecked the Table 1. results.
Reviewer 4 Report
Please see attachment.

As a non-native English speaker, it seems appropriate.
Author Response
Dear Reviewer,
thank You for an extensive review of the paper. The current study being examined is a pilot study, which means it is a preliminary investigation designed to test the feasibility, methods, and data collection procedures before conducting a more comprehensive and definitive research project. While the pilot study provides valuable insights, it usually involves a smaller sample size and is limited to specific locations, which might not fully represent the diversity and complexity of the target population. The future study will include a significantly larger number of participants compared to the pilot study and efforts will be made to recruit participants from various geographical locations, representing diverse communities, cultures, and backgrounds. This will help capture regional variations and ensure that the study's conclusions are not limited to a specific group but can be extended to a more diverse population
According to the comments we have corrected the paper and answered all comments.
Abstract:
1) Please shorten the abstract according to the author guidelines (maximum 200 words).
The abstract has been corrected according to the Instruction for Authors of Healthcare.
Background.Chest pain represents a prevalent complaint in emergency departments (EDs), where the precise differentiation between acute coronary syndrome and alternative conditions assumes paramount significance.This pilot study aimed to assess the HEART score implementation in West Balkan EDs. Methods.A retrospective analysis was performed on a prospective cohort comprising patients presenting with chest pain admitted to the EDs in Sarajevo, Zenica, and Belgrade between July and December 2022. Results.A total of 303 patients were included, with 128 classified as low-risk based on the HEART score and 175 as moderate-to high-risk.The low-risk patients exhibited younger age and a lower prevalence of cardiovascular risk factors.Laboratory and anamnestic findings revealed higher levels of C-reactive protein, ALT, creatinine, higher rates of moderately to highly suspicious chest pain history, greater number of cardiovascular risk factors and elevated troponin levels in moderate-to high-risk patients.Comparatively, among patients with a low HEART score, 2.3% experienced MACE, whereas those with a moderate-to high-risk HEART score had a MACE rate of 10.2%.The moderate- to high-risk HEART score demonstrated a sensitivity of 91.2% (95%CI 90.2%–93.4%) and specificity of 46.5% (95%CI 39.9%–48.3%) for predicting MACE. Conclusion.This pilot study offers preliminary insights into the integration of the HEART score within the emergency departments of the West Balkan region.
Keywords: words). https://www.mdpi.com/journal/healthcare/instructions (see Front
2) For better searchability on the Internet, use keywords that do not appear in the title or abstract. For example 1) chest pain = angina pectoris symptoms; ACS = myocardial infarction, risk score
We have corrected this point according to the recommendation.
Keywords: angina pectoris; myocardial infarction; HEART score; emergency department; risk score according to MESH Browser
Introduction:
3) "... in US emergency departments..." Do you have case numbers from your countries as well? Please add them.
Data regarding chest pain emergency department admission are scarce to none in West Balkan. So, there weren’t any official and unofficial information about it in our countries. One point of the further research is to investigate the current burden of it in our region.
4) Lines 52-58: “Although several historical features and laboratory results can assist in identifying patients with genuine ACS, none of them are precise enough
say acute coronary syndrome. When is an acute coronary syndrome present? In my opinion, when troponin is elevated. There are clear guidelines. I would the gold standard or a risk score? You should describe something here that in patients with unremarkable parameters, a risk score would be helpful to predict the risk of later developing ACS. That's probably the patients that matter in your study?
We have corrected it.
5) Line 73: Please add a point at the end of the sentence.
This has been added to the manuscript.
6) The authors presented different risk scores. I think it makes sense to tell the
reader why you chose the HEART Score. What are the advantages of the score?
You are welcome to put that in the last paragraph of the introduction.
7) Explain why you took the Heart Score. For example, the European Society of
Cardiology recommends the Grace Score for risk assessment.
In the assessment of chest pain patients, the HEART score exhibited higher discriminative capability in identifying individuals with and without Major Adverse Cardiac Events (MACE) when compared to the GRACE and TIMI scores. Additionally, the HEART score identified a substantial cohort of low-risk patients while maintaining a comparable level of safety in risk stratification.
We have added this point in the manuscript.
8) Lines 77-79: What does an intermediate risk or high risk say? In what time
frame can a cardiac event be expected?
We have added this point in the manuscript.
9) What other goals do you have with your pilot study? Which hypotheses do you
have?
We have correct it.
Material and Methods
10) Does the ethics vote approval include the other Balkan countries? Please complete this information.
The ethical approval includes only cities included in the study. Other Balkan countries will be regruted in the future study.
11) Please explain the meaning of “trained clinicians”. Are they physicians or other people who work in a clinic?
to be relied upon independently.”
You should explain what you mean when you
rephrase this sentence, because it doubts the gold standard.
Would you believe
We have corrected this point.
12) Subjects: Were women and men taken into account? Is the score also applicable for women? Were any pre-existing conditions raised that may affect the results? Have the included patients had a history of ACS?
Yes the score is applicable for men and women. Patients with previous heart conditions were also included as the score. In the section History highly suspicious anamnesis incorporates previous MI, CABG, PCI or other ischemic cardiac diseases.
13) Study instrument and data collection: Explain why you used a 30-day follow-up.
The 30-day follow-up period is considered relevant for capturing immediate post-acute events and complications that might occur shortly after an index event or intervention. It helps to understand the immediate impact of the treatment or intervention being studied.
Furthermore, using a longer follow-up period may introduce potential biases due to loss to follow-up, changes in treatment strategies, or unrelated events that could confound the study outcomes. A shorter follow-up period minimizes the risk of these potential confounding factors and ensures more focused and accurate data collection.
14) HEART score: I would recommend tabulating the contents of the Heart Score
influence on the laboratory parameters?
We tabulated it in Table 2.
15) Explain why you now form only 2 groups: low-risk and moderate to high-risk.
Patients with a moderate risk HEART score, admission for additional testing is recommended, while those with a high risk should be evaluated for potential invasive testing, while patients with low risk were discharged.
Results
16) Line 206 “angina”. It’s better to write “angina pectoris” because “angina” also means tonsillitis.
We have corrected this point.
Discussion
17) Lines 235-238: This is basic medical knowledge regarding the risk for ACS.
Do we need to add more data from the literature?
18) Lines 242-243: “These findings underscore the effectiveness of the HEART
or older? Women or men? Patients with or without pre-existing conditions? Specify this statement. It's too general. After all, you also write that they have predominantly studied men. Can you transfer this to women?
19) Please list the limitations of the study. No differentiation was made for gender, pre-existing conditions, medications. Age also does not play a role. It is a small sample to generalize. Are the results generalizable to women? Is the score appropriate for patients who are of a certain age.
We have listed the limitations. Age plays a role, as mentioned in the score calculation. Results are generalizable for women.
20) Discuss the advantages of the Heart Score over the Grace. There are many numbers of risk scores. Why should physicians use the Heart Score now? This needs to be worked out more.
We have added this point in the text.
21) You write in the title "a pilot study”. What other plans do you have? Will the study population be increased? What is the time frame you are planning for?
After this initial study we plan to recruit other emergency departments and do a major scale study in the region of West Balkan. Yes, the study population will be increased. We aim to investigate a 5-year long prospective analysis.
All these points have been added to the Discussion part.
Round 2
Reviewer 4 Report
See attachment.

See website.
Author Response
Dear Reviewer,
thank You very much for all the comments. As ordered we have corrected all the points mentioned in the paper.
According to the comments:
1. Please also add your answer to the manuscript so that readers will know. We have added this point in the manuscript.
2. Line 194: Please add a point at the end of the sentence. The point has been added.
3. Please also add your answer to the manuscript so that readers will know. We have added this point in the manuscript.
4. The former question 14 was not answered sufficiently. Only the first sentence was taken into account. Complete the information in the manuscript. We have added this point in the Methods section. Also we have added in the limitations section answers to the influence of age and medication on certain laboratory findings.
5. Explain why you now form only 2 groups: low-risk and moderate to high-risk. The former question 15: Complete the information in the manuscript. We have explained it in the Methods section.
7. You can also delete the sentence, as it is common knowledge. We have deleted it.
8. Question 18 was not answered. Please complete the answer in the discussion. One of the most important limitations was not considered. We have answered this question.
9. Perhaps you can add that the HeartScore is generalizable only to women with chest pain (thus typical symptoms). Since women also often have atypical symptoms, presumably falling through the screening here, there would be a gap in applicability. This point was added in the manuscript.
10. Do you also plan to look at gender in future surveys? If so, add it to the manuscript. Yes, we have added it in the text.
Thank You very much for the improvement of the initial paper.
Sincerely,
Dr Armin